# Unlearnable Examples: Making Personal Data Unexploitable

**Hanxun Huang[1]  Xingjun Ma[2]†  Sarah Monazam Erfani[1]  James Bailey[1]  Yisen Wang[3]†**
[1]The University of Melbourne, VIC, Australia
[2]Deakin University, Geelong, VIC, Australia
[3]Key Lab. of Machine Perception (MoE), School of EECS, Peking University, Beijing, China

## Abstract

The volume of "free" data on the internet has been key to the current success of deep learning. However, it also raises privacy concerns about the unauthorized exploitation of personal data for training commercial models. It is thus crucial to develop methods to prevent unauthorized data exploitation. This paper raises the question: *can data be made unlearnable for deep learning models?* We present a type of *error-minimizing* noise that can indeed make training examples unlearnable. Error-minimizing noise is intentionally generated to reduce the error of one or more of the training example(s) close to zero, which can trick the model into believing there is "nothing" to learn from these example(s). The noise is restricted to be imperceptible to human eyes, and thus does not affect normal data utility. We empirically verify the effectiveness of error-minimizing noise in both sample-wise and class-wise forms. We also demonstrate its flexibility under extensive experimental settings and practicability in a case study of face recognition. Our work establishes an important first step towards making personal data unexploitable to deep learning models. Code is available at https://github.com/HanxunH/Unlearnable-Examples.

## 1 Introduction

In recent years, deep learning has had groundbreaking successes in several fields, such as computer vision (He et al., 2016) and natural language processing (Devlin et al., 2018). This is partly attributed to the availability of large-scale datasets crawled freely from the Internet such as ImageNet (Russakovsky et al., 2015) and ReCoRD (Zhang et al., 2018b). Whilst these datasets provide a playground for developing deep learning models, a concerning fact is that some datasets were collected without mutual consent (Prabhu & Birhane, 2020). Personal data has also been unconsciously collected from the Internet and used for training commercial models (Hill, 2020). This has raised public concerns about the "free" exploration of personal data for unauthorized or even illegal purposes.

In this paper, we address this concern by introducing *unlearnable examples*, which aims at making training examples unusable for Deep Neural Networks (DNNs). In other words, DNNs trained on unlearnable examples will have a performance equivalent to random guessing on normal test examples. Compared with preserving an individual's privacy by obfuscating information from the dataset, what we aim to achieve here is different but more challenging. First, making an example unlearnable should not affect its quality for normal usage. For instance, an unlearnable "selfie" photo should be free from obvious visual defects so it can be used as a social profile picture. Ideally, this can be achieved by using imperceptible noise. In our setting, the noise can only be added to training examples on a single occasion (when the data is uploaded to the internet) prior to model training. However, DNNs are known to be robust to small noise either random (Fawzi et al., 2016) or adversarial (Szegedy et al., 2013; Goodfellow et al., 2014; Ma et al., 2018). It is still not clear whether small, imperceptible noise can stop the training of high-performance DNNs.

The development of unlearnable examples should take full advantage of the unique characteristics, and more importantly, the weaknesses of DNNs. One well-studied characteristic of DNNs is that they tend to capture more of the high-frequency components of the data (Wang et al., 2020a). Surprisingly, by exploiting this characteristic, we find that small random noise when applied in a class-wise

---

†Correspondence to: Xingjun Ma (daniel.ma@deakin.edu.au), Yisen Wang (yisen.wang@pku.edu.cn)

manner to the training data can easily fool DNNs to overfit to such noise (shown in Section 4). However, early stopping can effectively counteract this type of noise. DNNs are also known to be vulnerable to adversarial (or error-maximizing) noise, which are small perturbations crafted to maximize the model's error at the test time (Szegedy et al., 2013; Goodfellow et al., 2014). We find that error-maximizing noise cannot stop DNN learning when applied in a sample-wise manner to the training examples. This motivates us to explore the opposite direction to error-maximizing noise. Specifically, we propose a type of *error-minimizing noise* that can prevent the model from being penalized by the objective function during training, and thus can trick the model into believing there is "nothing" to learn from the example(s). We refer to an example that contains the error-minimizing noise as an *unlearnable example*. Error-minimizing noise can be generated in different forms: sample-wise and class-wise. Class-wise error-minimizing noise is superior to random noise and cannot be circumvented by early stopping. Sample-wise error-minimizing noise is the only effective noise that can make training examples unlearnable compared to random (Fawzi et al., 2016) or error-maximizing noise (Muñoz-González et al., 2017). Our main contributions are:

- We present a type of error-minimizing noise that can create unlearnable examples to prevent personal data from being freely exploited by deep learning models. The noise is small, imperceptible to human eyes, thus it does not reduce general data utility.
- We propose a bi-level optimization process to effectively generate different forms of error-minimizing noise: sample-wise and class-wise.
- We empirically verify the effectiveness and flexibility of error-minimizing noise for creating unlearnable examples. We also demonstrate the practical application of unlearnable examples in real-world scenarios via a case study on face recognition.

## 2 RELATED WORK

In this section, we briefly review most relevant works in data privacy, data poisoning, adversarial attacks against deep learning models.

**Data Privacy.** Privacy issues have been extensively studied in the field of privacy-preserving machine learning (Shokri & Shmatikov, 2015; Abadi et al., 2016; Phan et al., 2016; 2017; Shokri et al., 2017). While these works have made significant progress towards protecting data privacy, they are developed based on the assumption that the model can freely explore the training data and turn to protect the model from leaking sensitive information about the training data. In this paper, we consider a more challenging scenario where the goal of the defender is to make personal data completely unusable by unauthorized deep learning models. Fawkes (Shan et al., 2020) has made the first attempt towards this type of strict situation. By leveraging the targeted adversarial attack, Fawkes prevents unauthorized face tracker from tracking a person's identity. This work is similar to ours as we share a common objective that prevents unauthorized data usage. In contrast to the targeted adversarial attack, we propose a novel error-minimizing noise to produce unlearnable examples which can be used as a generic framework for a wide range of data protection tasks.

**Data Poisoning.** Data poisoning attacks aim to degrade the model's performance on clean examples by modifying the training examples. Previous work has demonstrated a poisoning attack on SVM (Biggio et al., 2012). Koh & Liang (2017) proposed to poison the most influential training examples using adversarial (error-maximizing) noise against DNNs, which has also been integrated into an end-to-end framework (Muñoz-González et al., 2017). Although data poisoning attacks can potentially prevent free data exploitation, these approaches are quite limited against DNNs and hard to operate in real-world scenarios. For example, poisoned examples can only slightly decrease DNNs' performance (Muñoz-González et al., 2017), and often appear distinguishable to clean examples (Yang et al., 2017) which will reduce normal data utility. The backdoor attack is another type of attack that poisons training data with a stealthy trigger pattern (Chen et al., 2017; Liu et al., 2020). However, the backdoor attack does not harm the model's performance on clean data (Chen et al., 2017; Shafahi et al., 2018; Barni et al., 2019; Liu et al., 2020; Zhao et al., 2020). Thus, it is not a valid method for data protection. Different from these works, we generate unlearnable examples with invisible noise to "bypass" the training of DNNs.

**Adversarial Attack.** It has been found that adversarial examples (or attacks) can fool DNNs at the test time (Szegedy et al., 2013; Goodfellow et al., 2014; Kurakin et al., 2016; Carlini & Wagner, 2017; Madry et al., 2018; Jiang et al., 2019; Wu et al., 2020a; Bai et al., 2020; Croce & Hein, 2020; Wang

et al., 2020b; Duan et al., 2020; Ma et al., 2020). The adversary finds an error-maximizing noise that maximizes the model's prediction error, and the noise can be crafted universally for the entire test set (Moosavi-Dezfooli et al., 2017). Adversarial training has been shown to be the most robust training strategy against error-maximizing noise (Madry et al., 2018; Zhang et al., 2019; Wang et al., 2019; Wu et al., 2020b; Wang et al., 2020c). Adversarial training can be formulated as a min-max optimization problem. In this paper, we explore the opposite direction of error-maximizing noise, i.e., finding small noise that minimizes the model's error via a *min-min* optimization process.

## 3 UNLEARNABLE EXAMPLES AND ERROR-MINIMIZING NOISE

### 3.1 PROBLEM STATEMENT

**Assumptions on Defender's Capability.** We assume the defender has full access to the portion of data which they want to make unlearnable. However, the defender cannot interfere with the training process and does not have access to the full training dataset. In other words, the defender can only transform their portion of data into unlearnable examples. Moreover, the defender cannot further modify their data once the unlearnable examples are created.

**Objectives.** We formulate the problem in the context of image classification with DNNs. Given a typical $K$-class classification task, we denote the clean training and test datasets as $\mathcal{D}_c$ and $\mathcal{D}_t$ respectively, and the classification DNN trained on $\mathcal{D}_c$ as $f_\theta$ where $\theta$ are the parameters of the network[*]. Our goal is to transform the training data $\mathcal{D}_c$ into unlearnable dataset $\mathcal{D}_u$ such that DNNs trained on the $\mathcal{D}_u$ will perform poorly on the test set $\mathcal{D}_t$.

Suppose the clean training dataset consists of $n$ clean examples, that is, $\mathcal{D}_c = \{(\boldsymbol{x}_i, y_i)\}_{i=1}^n$ with $\boldsymbol{x} \in \mathcal{X} \subset \mathbb{R}^d$ are the inputs and $y \in \mathcal{Y} = \{1, \cdots, K\}$ are the labels and $K$ is the total number of classes. We denote its unlearnable version by $\mathcal{D}_u = \{(\boldsymbol{x}_i', y_i)\}_{i=1}^n$, where $\boldsymbol{x}' = \boldsymbol{x} + \boldsymbol{\delta}$ is the unlearnable version of training example $\boldsymbol{x} \in \mathcal{D}_c$ and $\boldsymbol{\delta} \in \Delta \subset \mathbb{R}^d$ is the "invisible" noise that makes $\boldsymbol{x}$ unlearnable. The noise $\boldsymbol{\delta}$ is bounded by $\|\boldsymbol{\delta}\|_p \leq \epsilon$ with $\|\cdot\|_p$ is the $L_p$ norm, and $\epsilon$ is set to be small such that it does not affect the normal utility of the example.

In the typical case, the DNN model will be trained on $\mathcal{D}_c$ to learn the mapping from the input space to the label space: $f : \mathcal{X} \to \mathcal{Y}$. Our goal is to trick the model into learning a strong correlation between the noise and the labels: $f : \Delta \to \mathcal{Y}, \Delta \neq \mathcal{X}$, when trained on $\mathcal{D}_u$:

$$\arg\min_\theta \mathbb{E}_{(\boldsymbol{x}', y) \sim \mathcal{D}_u} \mathcal{L}(f(\boldsymbol{x}'), y) \tag{1}$$

where, $\mathcal{L}$ is the classification loss such as the commonly used cross entropy loss.

**Noise Form.** We propose two forms of noise: sample-wise and class-wise. For sample-wise noise, $\boldsymbol{x}_i' = \boldsymbol{x}_i + \boldsymbol{\delta}_i, \boldsymbol{\delta}_i \in \Delta_s = \{\boldsymbol{\delta}_1, \cdots, \boldsymbol{\delta}_n\}$, while for class-wise noise, $\boldsymbol{x}_i' = \boldsymbol{x}_i + \boldsymbol{\delta}_{y_i}, \boldsymbol{\delta}_{y_i} \in \Delta_c = \{\boldsymbol{\delta}_1, \cdots, \boldsymbol{\delta}_K\}$. Sample-wise noise needs to generate noise separately for each example. This may have more limited practicality. In contrast to sample-wise noise, a class of examples can be made unlearnable by the addition of class-wise noise, where all examples in the same class have the same noise added. As such, class-wise noise can be generated more efficiently and more flexibly in practical usage. However, we will see that class-wise noise may get more easily exposed.

### 3.2 GENERATING ERROR-MINIMIZING NOISE

Ideally, the noise should be generated on an additional dataset that is different from $\mathcal{D}_c$. This will involve a class-matching process to find the most appropriate class from the additional dataset for each class to protect in $\mathcal{D}_c$. For simplicity, here we define the noise generation process on $\mathcal{D}_c$ and will verify the effectiveness of using an additional dataset in the experiments. Given a clean example $\boldsymbol{x}$, we propose to generate the error-minimizing noise $\boldsymbol{\delta}$ for training input $\boldsymbol{x}$ by solving the following bi-level optimization problem:

$$\arg\min_\theta \mathbb{E}_{(\boldsymbol{x}, y) \sim \mathcal{D}_c} \left[ \min_{\boldsymbol{\delta}} \mathcal{L}(f'(\boldsymbol{x} + \boldsymbol{\delta}), y) \right] \quad \text{s.t.} \quad \|\boldsymbol{\delta}\|_p \leq \epsilon \tag{2}$$

where, $f'$ denotes the source model used for noise generation. Note that this is a min-min bi-level optimization problem: the inner minimization is a constrained optimization problem that finds the

---

[*]We omit the $\theta$ notation from $f_\theta$ without ambiguity in the rest of this paper.

$L_p$-norm bounded noise $\boldsymbol{\delta}$ that minimizes the model's classification loss, while the outer minimization problem finds the parameters $\theta$ that also minimize the model's classification loss.

Note that the above bi-level optimization has two components that optimize the same objective. In order to find effective noise $\boldsymbol{\delta}$ and unlearnable examples, the optimization steps for $\theta$ should be limited, compared to standard or adversarial training. Specifically, we optimize $\boldsymbol{\delta}$ over $\mathcal{D}_c$ after every $M$ steps of optimization of $\theta$. The entire bi-level optimization process is terminated once the error rate is lower than $\lambda$. The detailed training pipeline is described in Algorithm 1 in the Appendix.

**Sample-wise Generation.** We adopt the first-order optimization method PGD (Madry et al., 2018) to solve the constrained inner minimization problem as follows:

$$\boldsymbol{x}'_{t+1} = \Pi_\epsilon\left(\boldsymbol{x}'_t - \alpha \cdot \text{sign}(\nabla_{\boldsymbol{x}}\mathcal{L}(f'(\boldsymbol{x}'_t), y))\right) \tag{3}$$

where, $t$ is the current perturbation step ($T$ steps in total), $\nabla_{\boldsymbol{x}}\mathcal{L}(f(\boldsymbol{x}'_t), y)$ is the gradient of the loss with respect to the input, $\Pi$ is a projection function that clips the noise back to the $\epsilon$-ball around the original example $\boldsymbol{x}$ when it goes beyond, and $\alpha$ is the step size. The perturbation is iteratively applied for $T$ steps after each $M$ step of model training as we explained earlier. The final output is a unlearnable example $\boldsymbol{x}'$ and the generated error-minimizing noise is $\boldsymbol{\delta} = \boldsymbol{x}' - \boldsymbol{x}$.

**Class-wise Generation.** Class-wise noise $\Delta_c$ can be obtained by a cumulative perturbation on all examples in a given class. For each example in class $k$ at step $t$, it applies $\boldsymbol{\delta}_k$ to the original example $\boldsymbol{x}$ and follows Equation 3 to produce $\boldsymbol{x}'_{t+1}$. The $\boldsymbol{\delta}_k$ accumulates over every example for the corresponding class $k$ in the entire bi-level optimization process.

## 4 EXPERIMENTS

In this section, we first demonstrate the effectiveness in creating unlearnable examples using random noise, error-maximizing noise and our proposed error-minimizing noise in both sample-wise and class-wise forms. We further empirically verify the effectiveness of error-minimizing noise on 4 benchmark image datasets. We then conduct a set of stability and transferability analyses of the noise. Finally, we show effectiveness in real-world scenarios via a case study on face recognition. More analyses regarding the effectiveness of error-minimizing noise on small patches or a mixture of class-wise noise can be found in Appendix F.

According to previous studies in adversarial research, small $L_\infty$-bounded noise within $\|\boldsymbol{\delta}\|_\infty < \epsilon = 8/255$ on images are imperceptible to human observers. We consider the same constraint for all types and forms of the noise in our experiments, unless otherwise explicitly stated.

**Experimental Setting for Error-Minimizing Noise.** We apply error-minimizing noise to the training set of 4 commonly used image datasets: SVHN (Netzer et al., 2011), CIFAR-10, CIFAR-100 (Krizhevsky, 2009), and ImageNet subset (the first 100 classes) (Russakovsky et al., 2015). The experiments on the ImageNet subset are to confirm the effectiveness of high-resolution images. For all experiments, the ResNet-18 (RN-18) (He et al., 2016) is used as the source model $f'$ to generate the noise. We use 20% of the training dataset to generate the class-wise noise and the entire training dataset for the sample-wise noise except for ImageNet [†]. We transform the entire training dataset into the unlearnable datasets for experiments in section 4.1 and section 4.2. Different percentages of unlearnable examples are used for experiments in section 4.3. We train four different DNNs on the unlearnable training sets: VGG-11 (Simonyan & Zisserman, 2014), ResNet-18 (RN-18), ResNet-50 (RN-50) and DenseNet-121 (DN-121) (Huang et al., 2017). We also use clean training sets as a comparison. Detailed training configurations settings can be found in Appendix B. We evaluate the effectiveness of unlearnable examples by examining the model's accuracy on clean test examples, i.e., the lower the clean test accuracy the better the effectiveness.

**Experimental Setting for Random and Error-Maximizing Noise.** For random noise, we randomly sample the noise from $[-\epsilon, \epsilon]$ independently for each training example (eg. sample-wise) or each class (eg. class-wise). For error-maximizing (adversarial) noise, we generate the noise using PGD-20 attack (Madry et al., 2018) using a pre-trained ResNet-18 model on the training set. We generate sample-wise error-maximizing noise for each training example, and the class-wise noise based on 20% of the training set following the universal attack procedure in (Moosavi-Dezfooli et al., 2017).

---

[†]We use 20% of the first 100 class subset as the entire training set (due to the efficiency)

Both the random and error-maximizing noise are applied to the same amount of training examples as our error-minimizing noise.

## 4.1 COMPARISONS OF RANDOM, ERROR-MAXIMIZING AND ERROR-MINIMIZING NOISE

First, we examine an extreme case that applies different types of noise to the entire training set. Figure 1 illustrates the effectiveness of both sample-wise and class-wise noise. In the sample-wise case, the network is robust to random or error-maximizing noise. This is understandable since DNNs are known to be robust to small random (Fawzi et al., 2016) and error-maximizing noise (Muñoz-González et al., 2017; Madry et al., 2018; Wang et al., 2019). Surprisingly, when applied in a class-wise manner, both types of noise can prevent the network from learning useful information from the data, especially after the 15-th epoch. This reveals that DNNs are remarkably vulnerable to class-wise noise. While effective in the middle and later training stages, class-wise random noise can still be circumvented by early stopping (eg. at epoch 15). This is also the case for error-maximizing noise, although not as easy as random noise since the highest clean test accuracy under error-maximizing noise is only 50%. From the perspective of making data unexploitable, both random and error-maximizing noise are only partially effective. In comparison, our error-minimizing noise is more flexible. As shown in Figure 1, the error-minimizing noise can reduce the model's clean test accuracy to below 23% in both settings. Moreover, it remains effective across the entire training process.

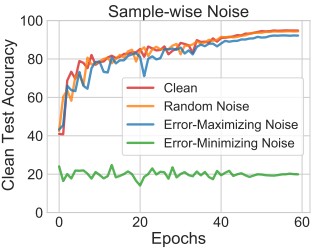 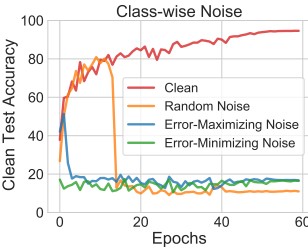

Figure 1: The unlearnable effectiveness of different types of noise: random, adversarial (error-maximizing) and our proposed error-minimizing noise on CIFAR-10 dataset. The lower the clean test accuracy the more effective of the noise.

The class-wise and sample-wise noises work in different ways. Class-wise noise has an explicit correlation with the label. Learning such correlation can effectively reduce the training error. Consequently, when there is class-wise noise, the model is tricked to learn the noise rather than the real content, reducing its generalization performance on clean data. The existence of class-wise noise only in the training data also breaks the i.i.d. assumption between the training and test data distribution. This also indicates that noises that can break the i.i.d. assumption might be effective techniques for data protection. However, in the sample-wise case, every sample has a different noise, and there is no explicit correlation between the noise and the label. In this case, only low-error samples can be ignored by the model, and normal and high-error examples have more positive impact on model learning than low-error examples. This makes error-minimizing noise more generic and effective in making data unlearnable.

## 4.2 EFFECTIVENESS OF ERROR-MINIMIZING NOISE ON DIFFERENT DATASETS

Table 1: The top-1 clean test accuracies (%) of DNNs trained on the clean training sets ($\mathcal{D}_c$) or their unlearnable ones ($\mathcal{D}_u$) made by sample-wise ($\Delta_s$) or class-wise ($\Delta_c$) error-minimizing noise.

| Noise Form | Model | SVHN | | CIFAR-10 | | CIFAR-100 | | ImageNet* | |
|---|---|---|---|---|---|---|---|---|---|
| | | $\mathcal{D}_c$ | $\mathcal{D}_u$ | $\mathcal{D}_c$ | $\mathcal{D}_u$ | $\mathcal{D}_c$ | $\mathcal{D}_u$ | $\mathcal{D}_c$ | $\mathcal{D}_u$ |
| $\Delta_s$ | VGG-11 | 95.38 | **35.91** | 91.27 | **29.00** | 67.67 | **17.71** | 48.66 | **11.38** |
| | RN-18 | 96.02 | **8.22** | 94.77 | **19.93** | 70.96 | **14.81** | 60.42 | **12.20** |
| | RN-50 | 95.97 | **7.66** | 94.42 | **18.89** | 71.32 | **12.19** | 61.58 | **11.12** |
| | DN-121 | 96.37 | **10.25** | 95.04 | **20.25** | 74.15 | **13.71** | 63.76 | **15.44** |
| $\Delta_c$ | VGG-11 | 95.29 | **23.44** | 91.57 | **16.93** | 67.89 | **7.13** | 71.38 | **2.30** |
| | RN-18 | 95.98 | **9.05** | 94.95 | **16.42** | 70.50 | **3.95** | 76.52 | **2.70** |
| | RN-50 | 96.25 | **8.94** | 94.37 | **13.45** | 70.48 | **3.80** | 79.68 | **2.70** |
| | DN-121 | 96.36 | **9.10** | 95.12 | **14.71** | 74.51 | **4.75** | 80.52 | **3.28** |

\* ImageNet subset of the first 100 classes.

Table 1 reports the effectiveness of both the sample-wise (eg. $\Delta_s$) and the class-wise (eg. $\Delta_c$) error-minimizing noise on datasets SVHN, CIFAR-10, CIFAR-100 and ImageNet subset. As shown in the table, our proposed method can reliably create unlearnable examples in both forms on all 4 datasets with images of different resolutions. Moreover, the noise generated on RN-18 works remarkably well to protect the data from other types of models. Compared to sample-wise noise, class-wise noise is particularly more effective, which can reduce the model's performance to a level that is close to random guessing. These results clearly show that error-minimizing noise is a promising technique for preventing unauthorized data exploration.

## 4.3 STABILITY ANALYSIS

We run a set of experiments to analyze the stability of error-minimizing noise in creating unlearnable examples, and answer two key questions regarding its practical usage: 1) *Is the noise still effective if only applied to a certain proportion or class of the data?* and 2) *Can the noise be removed by data augmentation or adversarial training?* Experiments are conducted on CIFAR-10 with RN-18.

**Different Unlearnable Percentages.** In practical scenarios, it is very likely that not all the training data need to be made unlearnable. For example, only a certain number of web users have decided to use this technique but not all users, or only a specific class of medical data should be kept unexploited. This motivates us to examine the effectiveness of error-minimizing noise when applied only on a proportion of randomly selected training examples. In other words, we make a certain percentage of the training data unlearnable while keeping the rest of the data clean. We train the model on this partially unlearnable and partially clean training set $\mathcal{D}_u + \mathcal{D}_c$. As a comparison, we also train the model on only the clean proportion, which is denoted by $\mathcal{D}_c$.

A quick glance at the $\mathcal{D}_u + \mathcal{D}_c$ results in Table 2 tells us that the effectiveness drops quickly when the data are not made 100% unlearnable. This is the case for both sample-wise and class-wise noise. The unlearnable effect is almost negligible even when the noise is applied to 40% of the data. Such a limitation against DNNs has also been identified in previous work for protecting face images using error-maximizing noise (Shan et al., 2020).

To better understand the above limitation, we take 80% as an example and plot the learning curves of the RN-18 model trained on 1) only the 20% clean proportion, 2) only the 80% unlearnable proportion, or 3) both. The results are shown in Figure 2 (a-b). Interestingly, we find that the 80% data with the error-minimizing noise are still unlearnable to the model, whereas the rest of the 20% clean data are sufficient for the model to achieve a good performance. In other words, models trained only on $\mathcal{D}_c$ demonstrate a similar performance as models trained on $\mathcal{D}_u + \mathcal{D}_c$. This phenomenon is consistent across different unlearnable percentages. This indicates that the high performance of the model on $< 100\%$ unlearnable dataset may not be a failure of the error-minimizing noise. We further verify this by investigating the scenario where only one class is made unlearnable.

Table 2: Effectiveness under different unlearnable percentages on CIFAR-10 with RN-18 model: lower clean accuracy indicates better effectiveness. $\mathcal{D}_u + \mathcal{D}_c$: a mix of unlearnable and clean data; $\mathcal{D}_c$: only the clean proportion of data. Percentage of unlearnable examples: $\frac{\mathcal{D}_u}{\mathcal{D}_c + \mathcal{D}_u}$.

| Noise Type | Percentage of unlearnable examples | | | | | | | | | |
|---|---|---|---|---|---|---|---|---|---|---|
| | 0% | 20% | | 40% | | 60% | | 80% | | 100% |
| | | $\mathcal{D}_u + \mathcal{D}_c$ | $\mathcal{D}_c$ | $\mathcal{D}_u + \mathcal{D}_c$ | $\mathcal{D}_c$ | $\mathcal{D}_u + \mathcal{D}_c$ | $\mathcal{D}_c$ | $\mathcal{D}_u + \mathcal{D}_c$ | $\mathcal{D}_c$ | |
| $\Delta_s$ | 94.95 | 94.38 | 93.75 | 93.10 | 92.56 | 91.90 | 89.77 | 86.85 | 84.30 | 19.93 |
| $\Delta_c$ | 94.95 | 94.24 | 93.75 | 92.99 | 92.56 | 91.10 | 89.77 | 87.23 | 84.30 | 16.42 |

**One Single Unlearnable Class.** We take the 'bird' class of CIFAR-10 as an example, and apply the error-minimizing noise (either sample-wise or class-wise) to all training images in the bird class. We train RN-18 on each of the unlearnable training set, and plot the prediction confusion matrix of the model on the clean test set in Figure 2 (c-d). The 'bird' class is indeed unlearnable when either the sample-wise or the class-wise error-minimizing noise is added to the class. Compared to sample-wise noise, class-wise noise is more effective with almost all the test 'bird' images being misclassified into other classes. Interestingly, this customized unlearnable class does not seem to influence much of the learning on other classes. Only the images from the unlearnable class are incorrectly predicted into other classes, not the other way around. This not only confirms that the unlearnable group of data

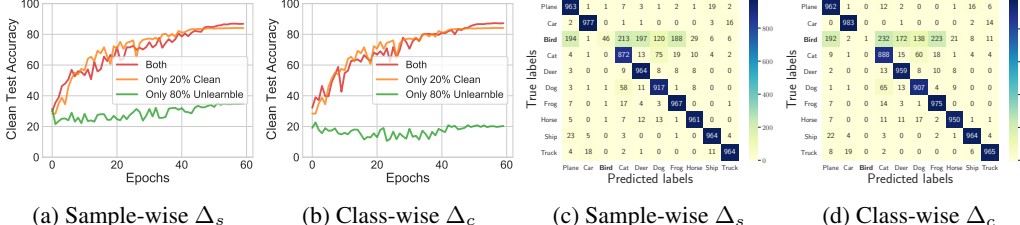

(a) Sample-wise $\Delta_s$    (b) Class-wise $\Delta_c$    (c) Sample-wise $\Delta_s$    (d) Class-wise $\Delta_c$

Figure 2: (a-b): For both sample-wise (a) and class-wise (b) noise, learning curves of RN-18 on CIFAR-10 dataset with different types of training data: 1) only 20% clean data, 2) only 80% unlearnable data, and 3) both clean and unlearnable data. (c-d): Prediction confusion matrices (on the clean test set) of two RN-18s trained on CIFAR-10 with the 'Bird' unlearnable class created by sample-wise (c) or class-wise (d) error-minimizing noise.

is indeed unlearnable to the model and it suggests that our error-minimizing noise can be flexibly applied to suit different protection tasks. A similar result can also be demonstrated on more than one unlearnable class (see Appendix C). In summary, if an individual can only apply the noise to portion of his/her data, these data will not contribute to model training. If an individual can apply the noise to all his/her data, the model will fail to recognize this particular class of data. In other words, our method is effective for the defender to protect his/her own data.

**Under Noise Filtering Techniques.** The resistance of our error-minimizing noise to 4 types of data augmentation techniques and adversarial training is analyzed in Appendix D. The noise is fairly robust to all 4 data augmentation techniques, and the highest accuracy that can achieve using data augmentation is 58.51% on CIFAR-10. Comparing with data augmentation, the error-minimizing noise is less resistant to adversarial training. With slightly increased $\epsilon$ on CIFAR-10, the noise can only compromise the model's clean test accuracy to 79%. Since adversarial training forces the model to learn only robust features (Ilyas et al., 2019), we believe our method can be improved by crafting the noise based on the robust features, which can be extracted from an adversarially pre-trained model on the clean data (Ilyas et al., 2019). We leave further explorations of these techniques as future work.

## 4.4 TRANSFERABILITY ANALYSIS

Another important question we haven't explored so far is that *Can error-minimizing noise be generated on a different dataset*? This is also important as a positive answer to the question increases the practicability of using unlearnable examples to protect millions of web users. We examine this capability for both sample-wise noise and class-wise error-minimizing noise.

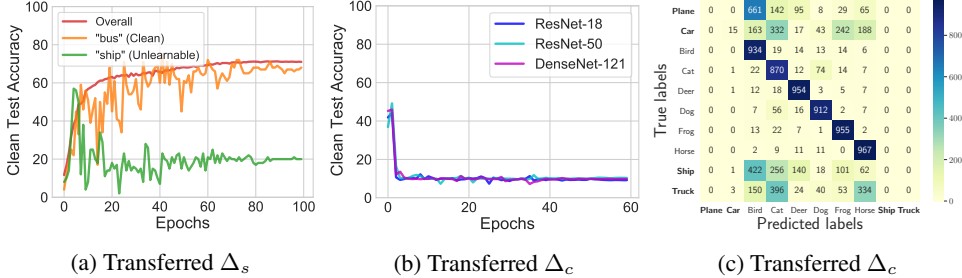

(a) Transferred $\Delta_s$    (b) Transferred $\Delta_c$    (c) Transferred $\Delta_c$

Figure 3: (a): Comparison between 'bus' (clean) class, 'ship' (unlearnable) class and the overall accuracy. (b): Clean test accuracy of RN-18/RN-50/DN-121 on unlearnable CIFAR-10 with error-minimizing noise crafted on ImageNet. (c): Prediction confusion matrix of RN-18 trained on CIFAR-10 with only 4 classes ('airplane', 'car', 'ship', 'truck') are unlearnable by ImageNet transferred noise, and the confusion matrix is computed on CIFAR-10 clean test set.

**Sample-wise Noise.** For $\Delta_s$, we generate the noise and create unlearnable 'ship' class examples on CIFAR-10 and add the unlearnable 'ship' class to CIFAR-100. For testing, we also include the clean test set of 'ship' class to the test set of CIFAR-100. Note the main difference of this experiment to the previous singe unlearnable class experiment is the unlearnable examples are generated on a different dataset. The class-wise and the overall accuracy on the clean test set are shown in Figure 3a. We

find that the 'ship' class is indeed unlearnable to the model at the end, despite a small amount (60% clean test accuracy) of information being learned in the early stage. We suspect this is because the 'ship' class shares some common features with other classes in CIFAR-100. Overall, the generated unlearnable examples can transfer to a different dataset. Note that this experiment simulates the scenario where a user uses a different dataset (eg. CIFAR-10) to generate the sample-wise noise to make his/her personal images unlearnable before uploading them to the Internet, and these images are then collected into a large CIFAR-100 dataset.

**Class-wise Noise.** For $\Delta_c$, we use the entire ImageNet as the source dataset to generate the noise using RN-18, then apply the noise to CIFAR-10. All ImageNet images are resized to $32 \times 32$. For each CIFAR-10 class, we determine its corresponding class in ImageNet based on the visual similarities between the classes. The detailed class mapping can be found in Table 4 in the Appendix. We train RN-18, RN-50 and DN-121 on the unlearnable CIFAR-10 created with the ImageNet transferred noise, and show their learning curves in Figure 3b. As can be observed, the transferred class-wise noise works reasonably well on CIFAR-10 with the clean test accuracy of all three models being reduced to around 10%. Compared to the noise directly generated on CIFAR-10 (see Figure 1), here the models can still capture some useful features in the first epoch. We conjecture this is because CIFAR-10 classes are more generic than ImageNet classes, and noise that minimizes the error of fine-grained classes can become less effective on generic classes. We also conduct an experiment on the ImageNet transferred noise to make 4 classes unlearnable of CIFAR-10. We plot the prediction confusion matrix of the RN-18 trained on the 4-class unlearnable CIFAR-10 in Figure 3c. It shows that the 4 classes are indeed unlearnable, and the model's performance on the other 6 classes is not affected. This confirms that transferred noise can also be used in customized scenarios.

## 4.5 REAL-WORLD SCENARIO: A CASE STUDY ON FACE RECOGNITION

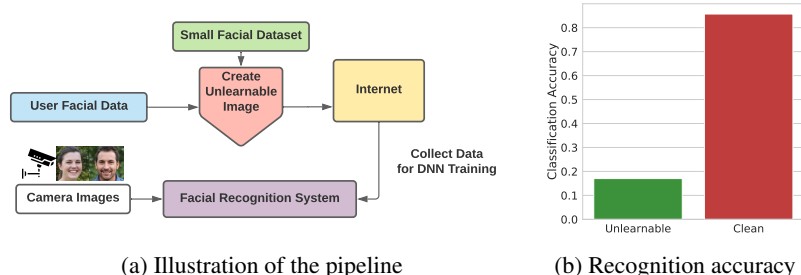

(a) Illustration of the pipeline      (b) Recognition accuracy

Figure 4: Preventing exploitation of face data using error-minimizing noise.

Here, we conduct a case study to apply error-minimizing noise to personal face images, which is arguably one most important real-world scenarios. In this scenario, the defender wants to prevent his/her face images from being exploited to train face recognition or verification systems. The setting is illustrated in Figure 4a. We assume the user has access to a small dataset of facial images and will use this small dataset to generate and apply the error-minimizing noise to his/her own face images before sharing them on online social media platforms. After this, the unlearnable version of face images on social media gets collected to train a DNN based facial recognition or verification system. The system will then be used to recognize one of the user's clean face image captured somewhere by a camera. The goal is to prevent the DNN from learning the defender's face images and make it perform poorly on his/her clean face images.

We conduct the experiments under two different settings: 1) *partially unlearnable* where a small subset of identities in the training set are unlearnable; or 2) *fully unlearnable* where the entire training set is unlearnable. We test our error-minimizing noise against both face recognition and verification models. Face recognition model classifies the identity (class) of a face image, while face verification model verifies whether two face images belong to the same identity. In the face verification problem, two face images are determined to be of the same identity if the cosine similarity between the features (extracted from a recognition model) of the two images is below a certain threshold.

**Experimental Settings.** We randomly split the WebFace dataset (Yi et al., 2014) into 80% for training and 20% for testing, according to each identity. In the partially unlearnable setting, we randomly select 50 identities from the training set of WebFace into a subset *WebFace-50* as the users who want to hide their identities. We also randomly select 100 identities from CelebA (Liu

et al., 2015) into a subset *CelebA-100* as the small dataset used to generate the unlearnable images for *WebFace-50*. The clean *WebFace-50* part of the WebFace training data will be replaced by its unlearnable version for model training. This partially unlearnable version of WebFace training set consists of 10,525 clean identities and 50 unlearnable identities. In the fully unlearnable setting, the entire WebFace is made unlearnable. In both settings, the error-minimizing noises are generated using RN-18 and *CelebA-100*, following the procedure described in Section 4. We train Inception-ResNet models (Szegedy et al., 2016) following a standard training procedure (Taigman et al., 2014; Parkhi et al., 2015).

**Effectiveness Against Face Recognition.** Here, we adopt the partially unlearnable setting. The accuracy of the Inception-ResNet model on the clean WebFace test set is reported in the Figure 4b, separately for the 50 unlearnable identities and the 10,525 clean identities. The result confirms that the 50 identities with the error-minimizing noise are indeed unlearnable and the recognition accuracy of their clean face images is only 16%, a much lower than the rest of the identities (86%).

**Effectiveness Against Face Verification.** Here, we adopt both the partially and the fully unlearnable settings. In the partially unlearnable setting, we evaluate the model performance on 6000 (3000 positives and 3000 negatives) face image pairs randomly sampled from the clean test set of WebFace. The pair is labeled as positive if the two face images are of the same identity, negative otherwise. In the fully unlearnable setting, we independently train a second Inception-ResNet model on the fully unlearnable WebFace. To further eliminate possible (dis)similarities shared across the images from the same dataset (e.g., WebFace), we follow the standard face verification evaluation protocol (Deng et al., 2019) and use the 6000 face pairs from LFW (Huang et al., 2008) for testing.

Figure 7 in Appendix G shows the Receiver Operating Characteristic (ROC) performance of the models trained in both settings. The model trained in the partially unlearnable setting still has a good verification performance on unlearnable identities, although the Area Under the Curve (AUC) is lower than the clean identities. This is because a small subset of unlearnable examples is not powerful enough to stop the learning of a feature extractor. In fact, 2,622 (approximately equal to 25% of WebFace) clean identities are sufficient to train a high-quality facial feature extractor (Parkhi et al., 2015). This indicates that, while it is easy to make data unlearnable to classification models, it is much more challenging to stop the learning of a feature extractor. Nevertheless, the proposed error-minimizing noise is still effective in the fully unlearnable setting, reducing the AUC to 0.5321 (the clean setting AUC is 0.9975). Whilst there are still many unexplored factors, we believe our proposed error-minimization noise introduces a new practical tool for protecting personal data from unauthorized exploitation.

## 5    CONCLUSION

In this paper, we have explored the possibility of using invisible noise to prevent data from being freely exploited by deep learning. Different from existing works, our method makes data *unlearnable* to deep learning models. The effectiveness of this approach could have a broad impact for both the public and the deep learning community. We propose a type of error-minimizing noise and empirically demonstrate its effectiveness in creating unlearnable examples. The noise is effective in different forms (eg. sample-wise or class-wise), sizes (eg. full image or small patch), and is resistant to common data filtering methods. It can also be customized for a certain proportion of the data, one single class or for multiple classes. Furthermore, the noise can be easily transferred from existing public datasets to make private datasets unlearnable. Finally, we verify the usefulness for real-world scenarios via a case study on face recognition. Our work opens up a new direction of preventing free exploitation of data. Although there are still many practical obstacles for a large-scale application of the error-minimizing noise, we believe this study establishes an important first step towards preventing personal data being freely exploited by deep learning.

### ACKNOWLEDGEMENT

Yisen Wang is partially supported by the National Natural Science Foundation of China under Grant 62006153, and CCF-Baidu Open Fund (OF2020002). This research was undertaken using the LIEF HPC-GPGPU Facility hosted at the University of Melbourne, which was established with the assistance of LIEF Grant LE170100200.

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

## A  ALGORITHM OF ERROR-MINIMIZATION GENERATION

---
**Algorithm 1** Error-minimizing Perturbations
---
1: **Input:** Source Model $\theta$, Perturbations $\boldsymbol{\delta}$, $L_p$ $\epsilon$, $(\boldsymbol{x}, y) \in D_c$, Stop Error $\lambda$, Optimization steps $M$
2: **Output:** $\delta$
3: **repeat**
4:     **for** $m$ **in** $1 \cdots M$ **do**
5:         $i, \boldsymbol{x}_i, y_i = \text{Next}(\boldsymbol{x}, y)$
6:         **if** sample-wise **then**
7:             $\boldsymbol{\delta} = \boldsymbol{\delta}_i$
8:         **else if** class-wise **then**
9:             $\boldsymbol{\delta} = \boldsymbol{\delta}_{y_i}$
10:        **end if**
11:        Optimize$(\boldsymbol{x}_i + \boldsymbol{\delta}, \boldsymbol{y_i}, \theta)$
12:     **end for**
13:     **for** $\boldsymbol{x}_j, y_j$ **in** $\boldsymbol{x}, y$ **do**
14:         $\boldsymbol{\delta} = \text{Perturbation}(\boldsymbol{x}_i, y_i, \theta, \boldsymbol{\delta})$                ▷ Follow Equation 3
15:         Cilp$(\boldsymbol{\delta}, -\epsilon, \epsilon)$
16:     **end for**
17:     $error = \text{Eval}(\boldsymbol{x}, y, \boldsymbol{\delta}, \theta)$
18: **until** $error < \lambda$
---

## B  MORE EXPERIMENT SETTING

For all experiments, we use $L_p$-norm with to regularize the imperceptibility, $\epsilon = 8/255$ for CIFAR and SVHN, $\epsilon = 16/255$ for ImageNet, different $\epsilon$ is used in Appendix D for additional understandings. The iterative steps $T$ for Equation 3 is set to 20 steps for sample-wise noise and 1 for class-wise, $\alpha$ is set to $\epsilon/10$. Since the class-wise noise is generated universally for each class, small iterative steps avoid overfitting to the specific example. For the class-wise experiments, we only use 20% of the training data $\mathcal{D}_c$ to generate the noise $\boldsymbol{\delta}$ and apply to entire training dataset $\mathcal{D}_c \rightarrow \mathcal{D}_u$. The stop condition error rate is $\lambda = 0.1$ for $\Delta_c$ and $\lambda = 0.01$ for $\Delta_s$. For SVHN and CIFAR-10, we set the $M = 10$ for CIFAR-10, $M = 20$ for CIFAR-100 and $M = 100$ for ImageNet in the mini-setting. For all models and experiments, we use the Stochastic Gradient Descent (SGD) (LeCun et al., 1998) optimizer with momentum 0.9, initial learning rate 0.025 and cosine scheduler (Loshchilov & Hutter, 2017) without the restart. We train all DNN models for 30 epochs on SVHN, 60 epochs on CIFAR-10, 100 epochs on CIFAR-100 and ImageNet. To generated fully unlearnable setting CIFAR-10 dataset, the computational cost is roughly 10% of the standard model training time for class-wise noise and 60% of standard model training time for sample-wise noise. Overall, the cost of generating unlearnable examples is far less than training a model on the data to be protected.

## C  STABILITY ANALYSIS

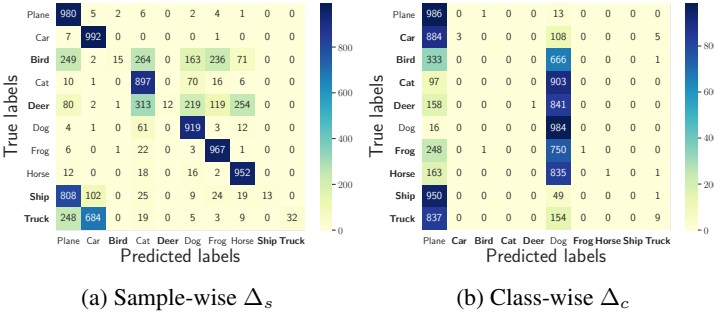

(a) Sample-wise $\Delta_s$                            (b) Class-wise $\Delta_c$

Figure 5: Prediction confusion matrix (on the clean test set) of two RN-18s trained on CIFAR-10 with the unlearnable classes in **bold** by (a) sample-wise or (b) class-wise error-minimizing noise.

## D    RESISTANCE TO DATA AUGMENTATION AND ADVERSARIAL TRAINING

**Resistance to Data Augmentation.** Another important property of the unlearnable examples created by error-minimizing noise is its resistance to data augmentations. Since standard data augmentation techniques like random shift, crop, flip and rotation have already been applied in previous experiments, here we consider 4 more advanced data augmentation techniques: Cutout (DeVries & Taylor, 2017), Mixup (Zhang et al., 2018a), Cutmix (Yun et al., 2019) and Fast Au-

Table 3: Clean test accuracy of RN-18 models trained on unlearnable CIFAR-10 with Cutout, Mixup, Cutmix and FA. Test accuracy of RN-18 trained on the clean CIFAR-10 is 94.95%.

| Noise | Cutout | Mixup | CutMix | FA |
|---|---|---|---|---|
| $\Delta_s$ | 19.30 | 58.51 | 22.40 | 42.70 |
| $\Delta_c$ | 14.62 | 17.63 | 16.19 | 22.89 |

toaugment (FA) (Lim et al., 2019). For Cutout, we set the cutout length to 16 pixels. For Mixup, we apply linear mixup of random pairs of training examples and their labels during the standard training process. For Cutmix, we apply linear mixup on the cutout region. For FA, we use the fixed augmentation policy, which consists of change contrast, brightness, sharpness, rotations and cutout. It is observed that advanced data augmentation techniques including Cutout, Mixup, Cutmix and FA can indeed remove the sample-wise noise to some extent, but far less effective on class-wise noise.

**Adversarial Training.** We also consider the Adversarial Training (AdvTrain), which is an augmentation based defence method against error-maximizing noise (Goodfellow et al., 2014; Madry et al., 2018). AdvTrain has been shown can effectively remove the non-robust features from the input and force the model to learn only the robust features (Ilyas et al., 2019). Compared to data augmentation techniques, adversarial training as a type of robust training technique is indeed more effective against both sample-wise and class-wise noise. However, adversarial training is known to suffer from a trade-off between robustness and accuracy (Zhang et al., 2019). For example, adversarial training only achieved 85% accuracy on the unlearnable CIFAR-10, there is still a roughly 10% (94.95% vs 85%) performance drop compared to standard training on the clean CIFAR-10. Moreover, due to the difficulty of min-max optimization, current adversarial training methods are still limited to small error-maximizing noise. To test this, we fix the maximum adversarial perturbation used by adversarial training to $8/255$, while increasing the perturbation of error-minimizing noise to $\epsilon = 24/255$. Note that adversarial training with perturbation $16/255$ or $24/255$ still suffers from convergence issues, even on small datasets like CIFAR-10. The results are shown in Figure 6. As can be confirmed, the clean test accuracy will drop to 79% on $\epsilon = 24/255$ error-minimizing noise.

The above results indicate that 1) error-minimizing noise is resistant to standard data filtering especially the class-wise noise; 2) although it is less resistant to adversarial training, the model's performance can still be significantly compromised by our error-minimizing noise noise. In future works, it is possible to develop more advanced unlearnable examples that can further decrease the performance of adversarial training. We believe our method can be improved by crafting the noise based on only the robust features, since adversarial training forces the model to learn only robust features (Ilyas et al., 2019).

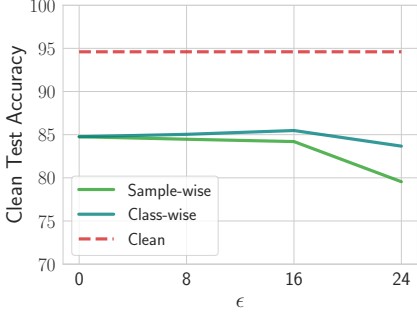

Figure 6: Clean test accuracy of adversarially trained RN-18 on unlearnable CIFAR-10 by different sizes ($\epsilon$) of error-minimizing noise, and the dashed line indicates the performance of RN-18 trained on clean CIFAR-10.

# E    CLASS-WISE NOISE TRANSFER FROM IMAGENET TO CIFAR-10

Table 4: Detailed class mapping from ImageNet classes to CIFAR-10 classes. This mapping is used for the second transfer experiment in Section 4.4.

| CIFAR-10 Class | ImageNet Class |
|---|---|
| Airplane | Airliner |
| Car | Wagon |
| Bird | Humming Bird |
| Cat | Siamese Cat |
| Deer | Ox |
| Dog | Golden Retriever |
| Frog | Tailed Frog |
| Horse | Zebra |
| Ship | Container Ship |
| Truck | Trailer Truck |

# F    FLEXIBILITY ANALYSIS OF ERROR-MINIMIZING NOISE

Here, we focus on the class-wise error-minimizing noise and further explore its flexibility from two perspectives: 1) effectiveness when applied to smaller patches rather than the entire image; and 2) effectiveness of two sets of mixed class-wise noise. These experiments are also conducted with RN-18 on the CIFAR-10 dataset. For all the error-minimizing noise, we fix the maximum perturbation to $\epsilon = 8/255$.

Table 5: Clean test accuracy of RN-18 models trained on unlearnable CIFAR-10 by either small patch noise applied at random location or a mixture of two type of noises.

| Methods | Patch Size | | | | Noise Mixture | |
|---|---|---|---|---|---|---|
| | $8 \times 8$ | $16 \times 16$ | $24 \times 24$ | $32 \times 32$ | $\Delta_{c1} \vee \Delta_{c2}$ | $\Delta_{c1} + \Delta_{u}$ |
| Standard Training | 87.23 | 19.19 | 26.42 | 16.42 | 25.41 | 14.94 |
| Fast Autoaugment | 90.66 | 36.69 | 50.60 | 22.89 | 56.20 | 30.40 |

**Effectiveness on Smaller Patches.** The unlearnable example can be very flexible and hard to be detected if the noise remains effective on smaller patches. To test this, we set the patch size to $32 \times 32$, $24 \times 24$, $16 \times 16$ and $8 \times 8$ (CIFAR-10 image size is $32 \times 32$). During the noise generation process when solving equation Equation 2, a random patch is selected and perturbed in each perturbation step and for each training example. Once a class-wise patch noise is generated, it then attached to a randomly selected location of a training example. The effectiveness of small patch noise is reported in Table 5. The effectiveness is still very high for patch size as small as $16 \times 16$, although it is indeed decreased on smaller patches. An interesting observation is that the FA augmentation becomes less effective on $16 \times 16$ noise than $24 \times 24$ noise. We suspect this is because smaller patch noise can more easily escape the augmentation operations and stays unchanged after the augmentation.

**Effectiveness of Mixed Noises.** We conduct this mix noise to simulate two possible real-world scenarios: 1) different users apply different noise apply to their own data; or 2) one set of noise is exposed and upgraded to a new set of noise. In this experiment, we apply the noise on the full image. We repeat the class-wise noise generation twice and generate two sets of noise: $\Delta_{c1}$ and $\Delta_{c2}$. For each CIFAR-10 training example, we randomly apply one of the above two class-wise noise to create unlearnable example. We note this mixture by $\Delta_{c1} \vee \Delta_{c2}$. Note that the class-wise mixture will eventually become the sample-wise noise if we keep mixing more class-wise noise sets. We also perform another mixture between $\Delta_{c1}$ and a random noise $\Delta_{u}$ sampled from $[-\epsilon, \epsilon]$ mixed by element-wise addition. We denote this mixture by $\Delta_{c1} + \Delta_{u}$. The effectiveness of mixed noise is also reported in Table 5. For both mixtures, the proposed error-minimizing noise remains highly effective, although there is a slight decrease. Surprisingly, the mixture of one class-wise noise with random noise is even more effective than the mixture of two class-wise noise, especially against the FA data augmentation. This makes the error-minimizing noise even more flexible in practical usage.

## G  RESULTS ON FACE VERIFICATION

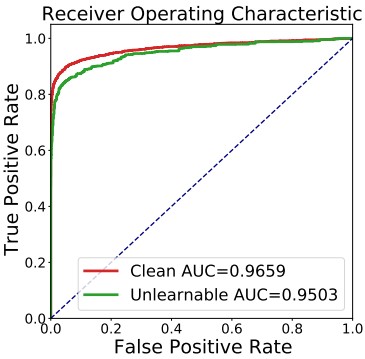
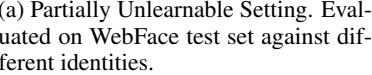

(a) Partially Unlearnable Setting. Evaluated on WebFace test set against different identities.

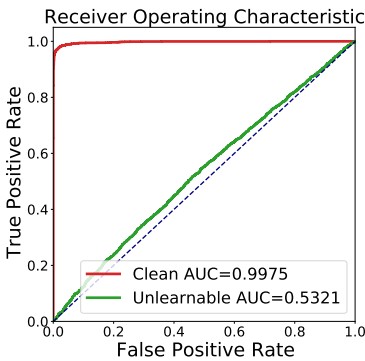

(b) Fully Unlearnable Setting. Evaluated on LFW using unlearnable and clean training data.

Figure 7: Results on face verification: the Receiver Operating Characteristic (ROC) of two Inception-ResNet models trained on partially unlearnable WebFace (left) and fully unlearnable WebFace (right). The ROC curves in the left plot are computed based on the clean WebFace test set while the ones in the right plot are computed based on LFW. Both Inception-ResNet models are trained as classifiers and tested as feature extractors. **Clean**: clean identities; **Unlearnable**: unlearnable identities. **Partially Unlearnable Setting:** 50 out of 10,575 identities are unlearnable.

## H  CLASS-WISE NOISE VS. BACKDOOR ATTACK

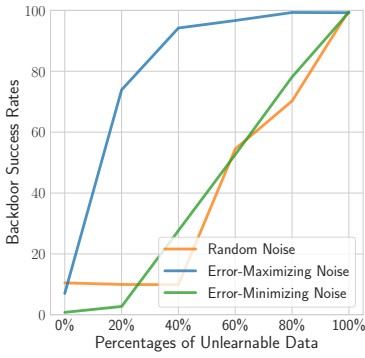

Figure 8: Backdoor attack success rates when 3 types of class-wise noises including random, error-maximizing and error-minimizing noise are applied as the backdoor triggers. The x-axis shows 6 RN-18 models trained on CIFAR-10 training set with different percentages of the data were poisoned by the class-wise noise. The y-axis shows the attack success rate: the percentage of all non-target class test images are predicted to the target class when attached with the target-class class-wise noise.

Here, we test if class-wise noise can be used as a trigger for the backdoor attack. Figure 8 shows the backdoor attack success rate when different types of class-wise noises are applied as the backdoor trigger. At the training time, we train RN-18 models on CIFAR-10 with 0%-100% of the training samples were made unlearnable by three types of class-wise noise: random noise, error-minimizing and our error-maximizing noise. At the test (attack) time, we compute the attack success rate as follows. There are 6 models for each type of class-wise corresponding the 6 unlearnable percentages. For each type of class-wise noise and each RN-18 model, we iteratively take a clean test image from the CIFAR-10 test set, then randomly select a target class (which is different from the true class of the test image). Then, we attach the target class class-wise noise to the clean test image. If the new test image is predicted by the model as the target class, then the attack is successful.

As shown in Figure 8, the class-wise error-maximizing (adversarial) noise is a highly effective trigger for backdoor attack, yet neither random nor our error-minimizing noise can be considered to be effective. The error-maximizing can achieve more than 75% attack success rate when only applied to 20% the training data. To achieve a similar level of attack success rate, both random and our error-minimizing noise should be added to at least 80% of the training data. This indicates that our method is different from backdoor attacks. The high attack success rate of adversarial noise may due to the fact that adversarial examples are hard (high error) examples, which force the model to pay more attention to the examples and remember more of the adversarial noise. Note that adversarial perturbations have also been used in backdoor research to enhance backdoor triggers (Turner et al., 2018; Zhao et al., 2020).

## I   DIFFERENT GENERALIZATION METHODS FOR ERROR-MINIMIZATION NOISE

The proposed error-minimizing noise is generated using PGD. Specifically, PGD is applied to solve the inner minimization problem in Equation 2. Since it is a typical constrained optimization problem, it can also be solved by other optimization (attack) methods such as FGSM (Goodfellow et al., 2014), L-BFGS (Szegedy et al., 2013) and C&W (Carlini & Wagner, 2017). However, the objective needs to be reformulated to apply some of these attacks. Here, we take L-BFGS (Szegedy et al., 2013) as an example to reformulate and solve the inner minimization problem in Equation 2. In (Szegedy et al., 2013), the adversarial attack problem is formulated as:

$$\text{minimize} \quad c\left\|\boldsymbol{\delta}\right\|_p + \mathcal{L}(\boldsymbol{x} + \boldsymbol{\delta}, y') \quad y' \neq y \tag{4}$$

where $c$ is a hyperparameter and the $\mathcal{L}$ is the objective function (e.g. the cross entropy adversarial loss). This form has been extended to other objective functions in C&W (Carlini & Wagner, 2017). The main objective of adversarial attack is to find small $L_p$ (i.e. $\left\|\cdot\right\|_p$) bounded noise $\boldsymbol{\delta}$ that can trick the model to output a wrong label $y' \neq y$. For unlearnable examples, we want to trick the model to predict the correct label $y$ with the highest confidence (i.e. lowest error). Following Equation 4, the inner minimization problem in Equation 2 can be reformulated as:

$$\text{minimize} \quad c\left\|\boldsymbol{\delta}\right\|_2^2 + \mathcal{L}(\boldsymbol{x} + \boldsymbol{\delta}, y). \tag{5}$$

We set $c = 1.0$ and use the cross entropy loss for $\mathcal{L}$. We apply L-BFGS to solve Equation 5 using Adam (Kingma & Ba, 2015) optimizer and generate sample-wise noise $\Delta_s$ on CIFAR-10. The number of optimization steps for Equation 5 is set to $T = 200$, while the number of optimization steps for the outer minimization in Equation 2 is set to $M = 100$ (see more details about $T$ and $M$ in Section 3.2). As shown in Figure 9, error-minimizing noise can also be generated using L-BFGS, and the noise is very effective in making the training examples unlearnable. However, compared to PGD, L-BFGS is slightly less effective.

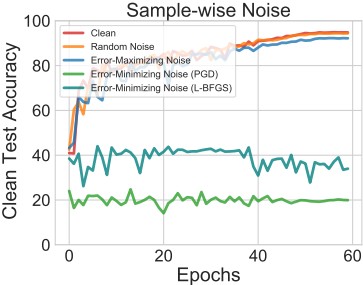

Figure 9: The unlearnable effectiveness of different types of noises on CIFAR-10 dataset: random, error-maximizing (adversarial), error-minimizing noise generated using PGD and error-minimizing noise generated using L-BFGS. The lower the clean test accuracy the more effective the noise in making training examples unlearnable. This is tested in the 100% unlearnable setting with the sample-wise noise.

