# OpenReview forum: "Unlearnable Examples: Making Personal Data Unexploitable"
_ICLR.cc/2021/Conference — ICLR 2021 Spotlight_

### Official Review · AnonReviewer3 · 2020-10-25
**This paper proposes a method to make personal data unexploitable. The objective is to introduce imperceptible noise to make the data robust against being automatically scraped and used to train commercial models. The idea is to make small sample-wise or class-wise perturbation in the direction that minimizes error and eventually makes the models trained on this model believe that there is nothing to learn from these samples.**

**Rating:** 7
**Confidence:** 4

**Review:**


$\textbf{Comments:}$
The paper's motivation is based on protecting private data and preventing its being scraped and used to train models. Even though motivation is clear and very important, the problem is the same as the works in crafting adversarial samples (i.e., the ones under data poisoning and adversarial attacks parts of the related work). The key difference is to apply Projected Gradient Descent (Mandry et al. 2018) in the reverse direction iteratively to *minimize* the loss function.  Furthermore, the performance evaluation will be the margin between models trained on completely clean data and sample-wise/class-wise adversarially corrupted data (in contrast to fooling a pretrained network in adversarial attack benchmarks).

$\bullet$ *Percentage of noisy training data:* In the "Assumptions on Defender's Capability" paragraph, the assumption is that only a part of the training set could be perturbed. The margin between error maximization and minimization on CIFAR-10 is remarkable (Figure 1), and this figure is misleading. 100\% of the training data was perturbed.

Besides, Table 2 gives accuracy in different ratios of noisy training samples. To understand whether perturbed training samples contribute to learning or not, I would compare them with clean training. For instance, in addition to the results of 20\% perturbed training setting in $\Delta_s$ and $\Delta_c$, training with only 80% of clean data without perturbed samples.

$\bullet$ *Comparison to PGD (Mandry et al. 2018):* Even under the class-wise perturbation, the noisy training data is learnable. In a sample-wise setting, "error-maximizing noise" is still learnable and performs very well; however, it performs around 20 and similar to "error-minimization" in a class-wise setting (Figure 1). If I am not wrong, Projected Gradient Descent, as proposed and applied in Mandry et al. 2018 (Figure 1, right side), reduces the performance the same as the proposed error minimization approach, and there is no performance gain.

$\bullet$ *Generalization to different Adversarial Attack methods:* Error minimization is shown using PDG only. There are several adversarial attack benchmarks on CIFAR-10 and ImageNet, such as CleverHans, Foolbox, or Realsafe (Considering different evaluation protocols, adopting these benchmarks for evaluation is a reasonable option to eliminate other factors). Is error minimization limited to only PDG or other methods? Did you try the effect of error minimization using any other method?

https://arxiv.org/abs/1707.04131

https://arxiv.org/abs/1610.00768

https://arxiv.org/abs/1912.11852

$\bullet$ *Different source-target models:* In all experiments, the source model is Resnet-18. The classification models used in performance evaluation are ResNet-18, ResNet-50 and DenseNet-121. All three models are based on residual blocks. In practice, we cannot assume the architecture that will be used by third-parties. Did you try completely different target models (such as AlexNet VGG,  Inceptionv3, etc.)

$\bullet$ *Application to face analysis:* Face recognition experiment is non-standard. I strongly recommend applying a standard dataset evaluation that would make comparisons possible. Both source and target sets are the datasets' subsets, and the selected identities might show visual (dis)similarities (i.e., ethnicity, age, gender). You can report the full performance on the entire target dataset (WebFace). Furthermore, face recognition models are trained as a recognition problem (with classification losses or metric learning) but tested in face verification settings (calculating the distance to query samples).  Reporting the distribution of these distances, for instance, Cumulative Matching Characteristic (CMC) and Receiver Operating Characteristic (ROC), would be more informative.

---

> ### Author Response · Authors · 2020-11-20
> **Response to AnonReviewer3**
>
> Thanks for your thoughtful comments. Please find our responses below to your questions.
>
> ---
> **Q1:** Figure 1 shows 100% perturbed data.
>
> **A1:** We would like to clarify that the assumption is “the defender has full access to the portion of data which they want to make unlearnable”. Figure 1 shows an extreme scenario where the defender (e.g., a data publisher) wants to make the entire dataset unlearnable. We have revised the description in Section 4.1 to make this clear.
>
> ---
>
> **Q2:** Table 2: in addition to the results of 20% perturbed training setting in Δs and Δc, training with only 80% of clean data without perturbed samples.
>
> **A2:** Thanks for the suggestion. This mixed 20% unlearnable and only 80% clean scenario was shown in Figure 2. In the updated version, we have included the clean only results for all percentages of unlearnable examples in Table 2. The results are consistent with previous findings.
>
> ---
> **Q3:** Figure 1: comparison to PGD.
>
> **A3:** Yes, PGD adversarial attack is also effective when generated and applied in a class-wise meaner. However, it doesn’ work in the sample-wise case (Figure1 left), like the random noise. This is one obvious limitation of adversarial noise when applied to making data unusable to deep learning models. PGD adversarial attacks have other capabilities, for example, fooling a DNN classifier at the test time, like you mentioned. For creating unlearnable data, class-wise noise and sample-wise noise work differently. Class-wise noise has an explicit correlation with the label. Learning such correlation can effectively reduce the training error. However, in the sample-wise case, every sample has a different noise, and there is no explicit correlation between the noise and the label. In this case, only low-error samples can be ignored by the model, and normal and high-error examples have a more positive impact on model learning than low-error examples. Compared to PGD (error-maximizing), our error-minimizing noise is more flexible for making data unexploitable. This is our main motivation and also the focus of this paper. We have added more discussion on this in Section 4.1.
>
>
> ---
> **Q4:** Generalization to different Adversarial Attack methods.
>
> **A4:** We agree that the inner problem defined in Eq. (2) is a typical constrained optimization problem that can be solved by different optimization (attack) methods such as L-BFGS, FGSM and PGD. PGD is a first-order gradient method, and is arguably the most reliable and efficient solver for constrained optimization problems. PGD adversarial attack is also known as the strongest first-order attack [1]. We will need to reformulate the objective to use some of these attacks. As a proof of concept example, we have reformulated the objective in the L-BFGS attack to create error-minimizing noise and experimented on sample-wise CIFAR-10. In comparison, the PGD is more effective compared with L-BFGS. See the added Appendix I for more details. We will explore more attacks in our future work.
>
> [1] Madry, Aleksander, et al. "Towards deep learning models resistant to adversarial attacks." ICLR, 2018.
>
> ---
> **Q5:** Did you try completely different target models (such as AlexNet VGG, Inceptionv3, etc.)?
>
> **A5:** Thanks for the suggestion. We have now included a VGG-11 target model in Table 1. VGG-11 is slightly more robust than other residual networks to the RN-18 generated error-minimizing noise on SVHN and CIFAR-10. It is similarly vulnerable to ResNets on CIFAR-100 and ImageNet datasets. Overall, our conclusions remain valid for VGG-11.
>
> ---
> **Q6:** Section 4.5: add face verification experiments and report Receiver Operating Characteristic (ROC).
>
> **A6:** Thanks for the suggestion. We have added the face verification experiments in Section 4.5 and the ROC plot in Appendix G. Overall, we find that it is still a challenge to make data unlearnable to feature extractors, especially when only a few (50 out of 10,575) identities (classes) are protected. This is because the rest of the identities are sufficient to train a good facial feature extraction model. Our method remains effective in the 100% unlearnable case.
>
> Our approach is focused on the classification problem rather than the representation learning problem, which is mentioned in Section 3.1 Objectives. Face verification is a representation learning problem. Our results show it is possible to make data unlearnable to classification models, but it is still a challenge to stop the learning of a feature extractor. We have added this discussion to make sure readers are aware of this limitation. We leave the unlearnable examples for representational learning in future works. Please find more of the results and analysis in the updated version of Section 4.5.
>
> ---

---

> > ### Comment · AnonReviewer3 · 2020-11-24
> > **Thanks for the answers and revisions.**
> >
> > My major concerns were generalizability to different attacks, generalizability under different target network architectures, and the performance/evaluation in face verification. Now I see all of them in the main text and appendix. Thus, I increase my rating to 7.

---

> ### Author Response · Authors · 2020-11-23
> **Any additional questions?**
>
> Dear AnonReviewer3,
> Thanks again for reviewing our paper. Please let us know if you have any additional questions or require further clarifications. We are happy to address them before the rebuttal ends.

---

### Official Review · AnonReviewer1 · 2020-10-27
**The idea presented in this paper is very interesting**

**Rating:** 8
**Confidence:** 4

**Review:**

Summary:
The authors proposed the idea of using invisible noise to make personal data unusable to authorized deep learning models. To achieve this goal, the authors proposed the idea of error-minimizing noise crafted by a min-min optimization method. The error-minimizing noise is then added to training examples to make them unlearnable to deep learning models. The idea is very well motivated and explained. The experiments not only confirm the exceptional effectiveness of the proposed method but also show its flexibility.

Pros:
1. The paper is very well written and easy to read.

2. I find the idea is very attractive and could have a significant social impact, especially considering the fact that personal data has already been overused without consent to train not just commercial but also illegitimate models to fake information or track people’s identity.

3. The idea of using the error-minimizing noise is well explained and the generation method is well formulated.

4. The experiments are very thorough, providing not only evidence of the superb effectiveness of the proposed noise over random or adversarial noise, but also the flexibilities and limitations of the proposed method. The real-world experiment makes the proposed idea even more convincing, although it is just a simple simulated scenario.

5. It seems that class-wise noise can easily break a classification model, which is somewhat interesting from the data protection perspective.

Cons:
1. I think the class-wise noise breaks the IID assumption of machine learning. It seems that breaking the essential assumptions in machine learning can break the model. Although this is not new, however, it turns out to be very interesting if used for data protection or similar ideas. The authors could have more discussions on this point. For example, what would happen if someone always used a different background (may be invisible) for each of the photos uploaded to social media, always shifting the newly collected test data to a different distribution. Can this serve as the same purpose?

2. The proposed noise seems not strong against adversarial training. Although adversarial training is costly and decreasing performance at this moment, they may be improved in the future. A discussion on the possible ways to generate the noise against adversarial training can be useful.

3. How the proposed method is related to backdoor attacks? It acts as a type of backdoor attack. Yes, backdoor attacks do not decrease the model’s performance on clean data. I think the “clean data” in the proposed setting should be the “poisoned data” rather than the ground truth clean data since both the training and testing data will be collected at the same time. I guess the only difference is that, in this protection setting, the defender cannot do anything about it unless recollecting or denoising the data, even if the defender finds the model is poisoned. I suggest the authors include more discussions around this point.

---

> ### Author Response · Authors · 2020-11-20
> **Response to AnonReviewer1**
>
> Thanks for your insightful comments and useful suggestions. Please find our responses below to your questions.
>
> ---
> **Q1:** Class-wise noise breaks the IID assumption.
>
> **A1:** Thanks for the insightful interpretation of our method. From the train-test data distribution perspective, class-wise noise indeed breaks the IID assumption: training data have label-correlated noise while test data are clean. If we understand correctly, you are suggesting to make the test examples always different to the training data by constantly changing the background. Our results in Figure 1 with the sample-wise random noise suggests that small random noise may not be effective enough to cause a test distribution shift. We believe this is an interesting direction for future work. We have also added the discussion to Section 4.1.
>
> ---
> **Q2:** Resistance against adversarial training (also mentioned by Reviewer #4).
>
> **A2:** We agree that small error-minimizing noise is not very effective against adversarial training, $\epsilon=8/255$ noise only decreases the clean accuracy by ~10% on CIFAR-10. We believe this can be improved in two different ways: 1) using large but semantically plausible noise; and 2) generating the noise on only robust features based on adversarially pre-trained models. We have updated this potential improvement in Appendix D. Please also find our response A2 to Reviewer #4.
>
> ---
> **Q3:** Relation to backdoor attack.
>
> **A3:** Thanks for the interesting insight. We agree that from the suggested perspective, our error-minimizing noise indeed behaves like backdoor attacks. To test this, we run a set of new experiments on CIFAR-10 with RN-18 and 3 types of class-wise noise in the standard backdoor setting. We find that both random class-wise noise and our error-minimizing noise are not effective backdoor triggers. This suggests that our method is different from backdoor attack. These results are reported in Appendix H.
>
> ---

---

> > ### Comment · AnonReviewer1 · 2020-11-24
> > **Thanks for addressing my concerns.**
> >
> > After reading the responses carefully, my previous concerns regarding this paper are addressed well. Thus, I increase my score to 8. Well done!

---

### Official Review · AnonReviewer2 · 2020-10-27
**Review 2 for "Unlearnable Examples: Making Personal Data Unexploitable "**

**Rating:** 7
**Confidence:** 4

**Review:**

Summary:
The authors studied the problem of data protection from a new perspective. They proposed one kind of error-minimizing noise to make the data (added noise) unlearnable. The noise is imperceptible to human eyes, and thus does not affect normal data utility. The idea is very interesting and inspiring. The authors conducted a series of solid experiments to validate the effectiveness of the proposed noise, and tested it on a real world task of face recognition.

Pros:
1. The idea of the paper is very interesting. Its motivation is intuitive and well explained. Considering adversarial training is to find the worst case example to make the training process robust, the authors proposed an opposite direction to find the easiest case to make the training process to learn nothing. The authors also proposed two types of noise: class-wise and sample-wise, which is a complete formulation.
2. The paper revealed an important problem to protect privacy, and proposed a simple yet effective method to prevent our data from unauthorized exploitation for training commercial models. I think it will attract a broad audience in the ICLR community.
3. The experiments are solid and comprehensive, considering the difference to random and error maximizing noises, effectiveness on different datasets and model architectures. The detailed stability and transferability analysis convince me why and how error minimizing noise works. Besides, they also show a real-world face recognition task to demonstrate its usefulness in practice.

Cons:
1. What is the overhead of generating and adding this kind of noise? The author did not mention it in the paper.
2. Revisiting Figure 1, I am curious to know why the sample-wise and class-wise noises perform so differently? especially for random and error-maximizing noise?
3. What is the difference between the proposed noise and the data poisoning methods?

---

> ### Author Response · Authors · 2020-11-20
> **Response to AnonReviewer2**
>
> Thanks for your valuable comments. Please find our responses below for your questions.
>
> ---
> **Q1:** What is the computational overhead of error minimizing noise?
>
> **A1:** In the 100% unlearnable setting on CIFAR-10 dataset, class-wise noise accumulates a 1-step sample-wise perturbation, while sample-wise noise takes 20 perturbation steps. The time it takes to generate the error-minimizing noise is roughly 10% of the standard model training time for class-wise noise and 60% of standard model training time for sample-wise noise. Overall, the cost of generating unlearnable examples is less than training a model on the data to be protected. We have added this discussion to Appendix B.
>
> ---
> **Q2:** In Figure 1, why do the sample-wise and class-wise noises perform so differently, especially for random and error-maximizing noise?
>
> **A2:** The class-wise and sample-wise noises work in different ways. Class-wise noise has an explicit correlation with the label. Learning such a correlation can effectively reduce the training error. Consequently, when there is class-wise noise, the model is tricked into learning the noise rather than the real content, reducing its generalization performance on clean data. However, in the sample-wise case, every sample has a different noise, and there is no explicit correlation between the noise and the label. In this case, only low-error samples can be ignored by the model, and normal and high-error examples have more impact on model learning than low-error examples. This makes error-minimizing noise more generic and effective in making data unlearnable. We have added this explanation to this in Section 4.1.
>
> ---
> **Q3:** What is the difference between the proposed noise and the data poisoning methods?
>
> **A3:** Our method might be regarded as a special type of data poisoning attack. The difference is that our method leverages the weakness of error minimization: the model learns only when there are errors. Previous data poisoning methods identify the most important examples to model learning and modify (e.g., change the labels or make large perturbations to) those examples to deteriorate model performance. Previous works have limited success for DNNs, require more $\epsilon$ budget or only reduce a little on the performance. Compared to these methods, our method is more flexible, as we have shown in the experiments in Section 4 and Figure 1. We have discussed the relationship between the proposed method and data poisoning in Section 2.
>
> ---

---

> > ### Comment · AnonReviewer2 · 2020-11-21
> > **Thank you**
> >
> > Thank the authors for proving the clarification. It is helpful that the authors added the discussion of sample-wise versus class-wise noise, and the comparison of data poisoning.
> >
> > This is a very interesting paper and the technical contribution is now more clear to me. I think it's an accept case.

---

### Official Review · AnonReviewer4 · 2020-10-28
**Unlearnable Examples: Making Personal Data Unexploitable.**

**Rating:** 7
**Confidence:** 3

**Review:**

This paper describes a method for making user data unusable for training machine learning models. It focuses primarily on image data. The basic idea is to use error-minimizing noise.

In this paper the author propose adding imperceptible to users error-minimizing noise that would make training data unusable for training. The authors proposed 2 methods for generating the noise: sample-wise and class-wise

This paper is well written. The code and the datasets used for the experimentation have been provided.

########################
Overall, I would recommend accepting this papers. My only concern is with the effectiveness of the proposed technique given what authors discussed in the appendix (see questions below).

The method was used on standard openly available image datasets. The results showed that when close to 100% of training samples have been updated with the error-minimizing noise the model performance went down considerably (as desired). However, when even 20% of training data was left clean model performance remained good.

########################
Questions:

From the appendix notes: it appears that adversarial training can significantly negate the effect of adding error-minimizing noise. The resulted model performance would be degraded when compared to model training using clean training data only but considering the fact that authors themselves acknowledged that user data with error-minimizing noise may just constitute a fraction of all the training data available for training a model the effectiveness of this technique may be limited.  (due to the effectiveness of adversarial training and the outsized influence of a relatively small number of clean data samples on model performance)
Can the authors discuss the issues with the effectiveness of their presented technique?


Mostly cosmetic:
page 5:  section 4.1 title “Error-maximizing” written twice.

---

> ### Author Response · Authors · 2020-11-20
> **Response to AnonReviewer4**
>
> Thanks for your thoughtful comments. We provide the following responses to your concerns.
>
> ---
> **Q1:** Limitations on the high unlearnable rate.
>
> **A1:** Although the noise has to be added to more than 80% of the training data to break the entire model, one only needs to add the noise to his/her own part of the data to make it unlearnable, as we showed in Figure 2, Figure 3 \(c\) and Figure 5. The goal is to make an individual's data unexploitable rather than break the DNN model. If an individual can only apply the noise to the portion of his/her data, these data will not contribute to model training. If an individual can apply the noise to all his/her data, the model will fail to recognize this particular class of data. In other words, our method is effective for the defender to protect his/her own data. However, it indeed requires more effort also to protect other people’s data. We have added this discussion to Section 4.3, One Single Unlearnable Class.
>
> ---
>
> ---
> **Q2:** Limitations against adversarial training
>
> **A2:** As we discussed in Appendix D, our method can indeed be mitigated by adversarial training when the noise is small. Our method can only drop ~10% of the test accuracy on CIFAR-10 at $\epsilon=8/255$, and by 16% when the noise size is increased to $\epsilon=24/255$. Since adversarial training encourages the model to learn more robust features [1], we believe our method can be improved by crafting the noise based on only the robust features, which can be extracted from an adversarially pre-trained model [1]. Note that adversarial training also comes with certain disadvantages (e.g. drop in accuracy and convergence issues on large perturbation), as we discussed in Appendix D.  We have updated this possible improvement in Section4.3 and Appendix D in the updated version.
>
> [1] Ilyas, Andrew, et al. "Adversarial examples are not bugs, they are features." Advances in Neural Information Processing Systems. 2019.

---

### Author Response · Authors · 2020-11-20
**Rebuttal Summary**

We sincerely thank all reviewers for their valuable comments and suggestions. We have made the following major updates during the rebuttal.

---
+ Table 1: added results for VGG-11 target model.
+ Table 2: added the results for using only the clean proportion of the training data.
+ Section 4.1: updated the discussion of the sample-wise and class-wise noise.
+ Section 4.3: updated the description for Table 2.
+ Section 4.5: added evaluation of our error-minimizing noise in the face verification setting.
+ Appendix D: updated possible improvement against adversarial training.
+ Appendix G: added ROC plots for section 4.5 face verification.
+ Appendix H: added new empirical results to show the difference between our method and backdoor attacks.
+ Appendix I: Generalization for the noise generation.

---

We have revised our paper according to all the valuable comments and please let us know if there is anything still not clear or any other suggestions.

---

### Decision · Program_Chairs · 2021-01-07
**Final Decision**

**Decision:**

Accept (Spotlight)

**Comment:**

The paper proposed a *novel* methodology for protecting personal data from unauthorized exploitation for training commercial models. The proposal is conceptually *intuitive* and technically *motivated*. It goes to the opposite direction of adversarial training: by adding certain error-minimizing noise (rather than error-maximizing noise) to the data, the model is fooled and believes there is nothing to learn from the data, and thus this can protect the data from being used for training. The paper is of not only *high quality* but also *broad interest* given the current social concerns about personal data privacy. I think its potential impact should get it a spotlight presentation.